# Design of High-Relaxivity Polyelectrolyte Nanocapsules Based on Citrate Complexes of Gadolinium(III) of Unusual Composition

**DOI:** 10.3390/ijms222111590

**Published:** 2021-10-27

**Authors:** Evgenia Burilova, Alexander Solodov, Julia Shayimova, Julia Zhuravleva, Darya Shurtakova, Vladimir Evtjugin, Elena Zhiltsova, Lucia Zakharova, Ruslan Kashapov, Rustem Amirov

**Affiliations:** 1A.E. Arbuzov Institute of Organic and Physical Chemistry, FRC Kazan Scientific Center of RAS, Arbuzov Street 8, 420088 Kazan, Russia; zhiltsova@iopc.ru (E.Z.); luciaz@mail.ru (L.Z.); rusl701@yandex.ru (R.K.); 2A.M. Butlerov Chemical Institute, Kazan Federal University, Kremlevskaya Street 18, 420008 Kazan, Russia; sanya.solodiv@live.com (A.S.); julia_shayimova@mail.ru (J.S.); yulialab6@mail.ru (J.Z.); darja-shurtakva@mail.ru (D.S.); vevtugyn@gmail.com (V.E.); rramirov58@mail.ru (R.A.)

**Keywords:** gadolinium(III), NMR-relaxation, polyelectrolytes solutions, citrate complexes, TEM

## Abstract

Through nuclear magnetic relaxation and pH-metry, the details of the complexation of gadolinium(III) ions with citric acid (H_4_L) in water and aqueous solutions of cationic polyelectrolytes are established. It is shown that the presence of poly(ethylene imine) (PEI) in solution affects magnetic relaxation behavior of gadolinium(III) complexes with citric acid (Cit) to a greater extent than polydiallyldimethylammonium chloride (PDDC). A large increase in relaxivity (up to 50 mM^−1^s^−1^) in the broad pH range (4–8) is revealed for the gadolinium(III)–citric acid–PEI system, which is particularly strong in the case of PEI with the molecular weight of 25 and 60 kDa. In weakly acidic medium (pH 3–7), the presence of PEI results in the formation of two tris-ligand associates [Gd(H_2_L)_3_]^3^^−^ and [Gd(H_2_L)_2_(HL)]^4^^−^, which do not exist in aqueous medium. In weakly alkaline medium (pH 7–10), formation of ternary complexes Gd(III)–Cit–PEI with the Gd(III)–to–Cit ratio of 1:2 is evidenced. Using transmission electron microscopy (TEM) and dynamic light scattering techniques (DLS), the formation of the particles with the size of 50–100 nm possessing narrow molecular-mass distribution (PDI 0.08) is determined in the solution containing associate of PEI with tris-ligand complex [Gd(H_2_L)_2_(HL)]^4^^−^.

## 1. Introduction

Investigation of complexation reactions of metal ions in polymer solutions is of great practical importance for ecology, medicine, and other fields of science. Polymer-enhanced or polymer-assisted ultrafiltration (PEUF or PAUF) is one of the trends in ecological and green materials, in which the data on the interactions between metal complexes and polyelectrolytes can be used [1,2,3,4]. PEUF is based on the formation of stable complexes of water-soluble polymers with heavy metal cations, which are separated from low-molecular-weight components of solution through filtration using chemically inert solid membranes. To bind metal cations, anionic polymers are usually employed (polyacrylic acid, polystyrene sulfonate, etc.) [1,5,6,7]. In this case, both direct coordination of functional groups with the cations of Cu(II), Ni(II), Zn(II), and Pb(II) and electrostatic binding of anionic metal complexes with protonated or quaternized groups of polyelectrolytes, as well as combination of the mentioned approaches are possible for polyamines. The latter variant was demonstrated on the example of the reaction of PEI with copper(II) through the formation of coordination bonds with nitrogen atoms of amino groups and simultaneously with the [Cr(CN)_5_(NO)]^3−^ complex, which is electrostatically attracted to protonated ammonium groups of polymer [8].

Such complexing agents as polyaminocarboxylates (IDA, NTA, EDTA) or citric acid can be present in the solutions for ultrafiltration [7,9,10,11,12], which could be added to wash metals out from specimens (soil, ores, etc.). Cationic polymers in the solution of mentioned ligands usually retain the complexes of the same composition (metal-to-ligand ratio of 1:1), which predominate in water at the same pH and concentrations of reagents. A drastic decrease in the effectiveness of binding upon addition of inorganic salts or excess of anions of ligand was mentioned [13,14]. The latter effect is caused by the fact that organic anions could bind cationic groups of polyelectrolytes [15,16] competing with metal complex. In this case, hydrogen bond formation between organic anion and functional groups of the polymer occurs together with electrostatic attraction.

The PEUF method was mainly developed towards double-charged 3d-metal ions, Cd(II), Pb(II), and U(VI), whereas lanthanides were employed quite rarely [17,18]. Meanwhile, investigation of the conditions of lanthanide isolation and separation is still relevant due to the broad application of lanthanides in the development of luminescent and magnetic materials, in medical diagnostics, and other directions. A number of lanthanide ions are paramagnetic; therefore, radio spectroscopy techniques can be employed for their study (EPR, NMR). Gd(III) is of high paramagnetism, which could be employed in its derivatives as the tags for bio-objects and, in particular, as contrast agents (CAs) in magnetic resonance imaging (MRI), which is studied in most detail [19,20,21]. NMR relaxation measurement is highly informative tool not only for the investigation of paramagnetic cation complexes (including development of new types of CAs for MRI), but also for the study of nanosized objects in solutions. Features of the mechanism of proton relaxation in gadolinium solutions (an increase in the relaxivity due to the slow-down of rotation of Gd(III) ions bound to nanoparticle) revealed not only aggregation in the solutions of surfactants [22,23] or lipophilic macrocycles [24,25], but also the binding of cations with polymers [26], or graphene oxide [27,28]. In addition, the authors of this current paper have previously discovered that metal ion complexes can form associates with oppositely charged ions of surfactants, macrocycles, or polymers [29,30,31,32].

It is usually considered in the methods for isolation and separation of metal ions (PEUF, ion exchange, etc.) that their complexes participate in sorption, which exist in water at the same concentrations as reagents and pH of the medium. However, polymers in solution (and even crosslinked resins in ion exchangers) could alter the composition of adsorbed complexes due to the shift of the complexation equilibrium, in analogy to micelles of surfactants [33,34]. In this case, both adverse effect (competition) and positive effect (synergy) are possible. One example is that apparent strengthening of the complexes of bis- and tris-iron(III) complexes with 4,5-dihydroxybenzene-1,3-disulfonic acid (Tiron) in solution was achieved upon introduction of the suspension of fine-dispersed (with the particle size of less than 30 μm) Amberlyt A-27 anionite [35]. Existing examples indicate that charged polymers or surfactant micelles can favor the formation of known complexes in solution. However, there are few data on the stabilization of the compounds of different composition, the formation of which is either difficult in solution or was not recorded. For example, the formation of uncharged gadolinium tris-salicylate in acidic medium was established due to its solubilization by nonionic micelles [36]. The same could occur in the case of a polyelectrolyte, if the complex possesses a very high charge and, consequently, binds the macromolecule effectively. Therefore, this study attempted to change the existence regions in solution and even the composition of citrate complexes of gadolinium in the presence of cationic polymers using the mentioned potential of NMR relaxation method.

This study is also important in context of the preparation of high-relaxivity compositions based on metal complexes as the models of high-performance of CAs for MRI. Commercial CAs represent the compounds of Gd(III) with polyaminocarboxylates (DTPA, DOTA, etc.) [20,37], which usually possess relaxivity values of 4–5 mM^−1^s^−1^ (20 MHz, 37 °C). An increase in relaxivity (which is equivalent to the increase in the contrast of MRI images) could improve the diagnostic accuracy and decrease the dosage of CAs, which are nephrotoxic [38]. Employment of the compositions, which provide binding of gadolinium with macromolecular ligands or utilize supramolecular interactions [39,40,41,42,43,44], including those with poly(ethylene imine) [45], is a promising approach to the increase in the relaxivity of CAs. The authors of this paper previously analyzed the potential of the design of such compositions based on Mn(II) complexes with DTPA in PEI solutions [32]; however, high relaxivity was observed in the solutions beyond physiological range and the compositions decomposed in the presence of 150 mM NaCl. Thus, it is important to analyze the relaxation behavior of various gadolinium complexes in the presence of polymers for the preparation of the compositions possessing high relaxivity and stability against salts.

In this work, transverse NMR relaxation at 19.65 MHz and potentiometric titration techniques were employed for comparative study of the interaction of gadolinium with citric acid in water and polymer solutions represented by branched PEI of various molecular masses and quaternized polyelectrolyte, PDDC. The former is a cationic polymer in acidic medium (as a result of protonation of amino groups) and partially cationic in neutral medium, whereas the molecules of PDDC are positively charged in the entire pH range. DLS and TEM techniques were employed for the measurement of the particle size in the prepared compositions.

## 2. Results and Discussion

### 2.1. Formation of Gadolinium(III) Citrate Complexes in Water

Figure 1 shows the pH dependences of relaxation effectiveness values of gadolinium(III) in water, PEI solution, and citric acid solution.

The relaxation curve of gadolinium aqua-ions [Gd(H_2_O)_8_]^3+^ in water (Figure 1) preserves constant R_2_ values up to neutral medium (15.5 ± 0.5 mM^−1^s^−1^) and there is a decay of relaxivity after pH 6, which is caused by a gradual hydrolysis of gadolinium(III) resulting in the precipitation of insoluble hydroxide [Gd(OH)_3_]. The presence of cationic form of PEI in gadolinium solution does not change the state of its aqua-ions (it possesses the same R_2_ values up to pH 6) and there is also a decay of the curve at pH > 6. However, the limiting value of the relaxivity curve for PEI solution is much higher than that of aqueous solution of Gd(III) at the same pH (4 vs. 0.4 mM^−1^s^−1^). A similar effect was previously discovered by the authors of this paper not only for Gd(III), but also for Mn(II) [26]. The aforementioned phenomenon could not be caused by the binding of the hydrolyzed form of metal due to the competition of OH^-^-ions and amino groups of the branched PEI molecule for binding with metal ions [46], even though gadolinium ions do not form complexes in solutions with ammonia, ethylene diamine, and other low-molecular amines.

The decay of R_2_ values starts at pH 4 in the solution with citric acid (Figure 1). Such decrease in relaxivity before hydrolysis of gadolinium ions is caused by the substitution of functional groups of ligands for water molecules, which results in the decrease in the relaxation effectiveness of the protons of water molecules in the first sphere of paramagnetic ions. In this case, the R_2_ values change stepwise, which indicates a successive formation of a series of citrate complexes of gadolinium. Derived from the fact that relaxivity in solution corresponds to the R_2_ value for gadolinium aqua-ion up to pH 4, it can be suggested that there is no interaction with citric acid. However, this fact contradicts to the results of pH-metric titration of aqueous solution of citric acid containing gadolinium (Figure 2).

Titration curves of citric acid with and without gadolinium become separated at first points, which unambiguously shows the formation of citrate complexes in acidic medium (pH 3). The results of titration were modeled using a CPESSP computer program for following equilibria:xGd^3+^ + yH_4_L ⇆ Gd_x_L_y_H_z_^z+3x−4y^ + (4y − z)H^+^(1)

Table 1 shows the compositions of the complexes and equilibrium constants of their formation and stability for following equilibria:xGd^3+^ + yH_z/y_L^z−4y^ ⇆ Gd_x_L_y_H_z_^z+3x−4y^(2)

The determined values of stepwise dissociation constants of citric acid are sufficiently close to known values (pK_a1_ = 3.13, pK_a2_ = 4.76 and pK_a3_ = 6.40 [47] The model, including the complexes revealed by potentiometric data, was used for the modeling of NMR relaxation data given in Figure 1. Optimized relaxivity values of all complexes are also given in Table 1. The complexes [GdH_2_L]^+^ and [GdHL]^0^, which are formed in solutions at pH less than 4, possess the relaxivity values, which nearly coincide with the R_2_ value for gadolinium aqua-ion; therefore, their values were fixed. Thus, there is a rare case in this system, when replacement of water molecules by the ligand in aqua-ion [Gd(H_2_O)_8_]^3+^ upon complex formation hardly alters relaxivity. The anticipated decrease in relaxivity as a result of substitution of ligand for water molecules upon formation of [GdH_2_L]^+^ and [GdHL]^0^ complexes is presumably compensated by the change of other parameters, which affect the rate of relaxation (correlation times τ_c_, the distance from unshared electrons to protons, and others) [19,48]. When there is no excess of the ligand at pH > 4, two other complexes with the 1:1 metal-to-ligand ratio are successively formed, for which the relaxivity values differ significantly from the value for gadolinium aqua-ions. Two complexes, [GdHL]^0^ and [GdL]^−^, were previously identified in [49], while the authors from [50] showed the formation of four complexes, [GdH_2_L]^+^, [GdHL]^0^, [GdL]^−^, and [GdL(OH)]^2^^−^. 

In contrast to polydentate DOTA or DTPA complexing agents which can occupy up to eight positions in the inner sphere, one molecule of citric acid cannot saturate the coordination environment of one gadolinium ion. Therefore, citrate bis-complexes and polynuclear compounds are known [51]. Gadolinium and some other lanthanides are extracted by macromolecular amines from citric acid solutions in the form of their bis-citrate complexes 3[RNH_3_)]^+^[Ln(HL)_2_]^3^^−^ [52]. The possibility of the formation of tris-complex [Ln(H_3_L)_3_]^0^ for some lanthanides (Ln) was reported earlier from the ion exchange data [53]; however, more recent works did not support this finding [51].

At excess of ligand, the formation of bis-ligand complexes was revealed from the results of titration (Table 1). In this case, the pH ranges of the existence of bis-ligand complexes [Gd(HL)_2_]^3^^−^ and [GdL_2_]^5^^−^ determined pH-metrically, correspond to the stepwise decays of relaxivity, which were discovered at the excess of citric acid (Figure 1). At a ten-fold excess of the ligand, the relaxation form is the same as that at a three-fold excess. This fact indicates that the complexation pattern does not change at a large excess of citric acid.

There is yet no unambiguous opinion on the participation of a hydroxyl group of citric acid in coordination with metal cations. Meanwhile, the problem of the coordination of citrate ions in complexes is important, because it could further rationalize whether there are uncoordinated carboxylate ions in the complexes, which could electrostatically attract the cationic fragments of polyelectrolyte. It should be noted that the authors from [50] insist on the coordination of citrate ions only through carboxylic groups including the complexes formed in alkaline solutions. Meanwhile, according to X-ray diffractometry data, citrate ion is linked to gadolinium cation through five- and six-membered cycles with its own hydroxyl group and two carboxylic groups in the [GdHL(H_2_O)_2_·H_2_O] complex [54]. C-13 NMR data show that such coordination is most probable for monoligand citrate complexes of gadolinium in solution [49]. An analogous statement was made regarding the bis-citrate complex of chromium(III) [Cr(HL)_2_]^3^^−^ possessing the same composition as [Gd(HL)_2_]^3^^−^ [55]. Favorable proton elimination from the hydroxylic group of citric acid as compared with the proton release from coordinated water in the [GdHL]^0^ and [GdL]^−^ complexes was mentioned in [49]. The authors of this current study previously showed a decisive role of the hydroxy group of citric acid in the formation of lanthanide complexes and their diversity (formation of homo- and heteropolynuclear compounds) in the example of comparative complexation of citric and tricarballylic acid (1,2,3-Propanetricarboxylic acid) [56].

### 2.2. Effect of PEI on the Complexation of Gadolinium(III) Ions with Citric Acid

As mentioned above, there are no studies on the complexation of lanthanides with citric acid in solutions of cationic polymers, whereas PEUF was carried out using PEI only towards some cations and without addition of ligands [17]. PEUF was investigated in most detail for the retention of citrate complexes of metals using cationic polymers (PEI, PDDC, chitosan) with regard of copper(II) [10,11,13,14]. In the case of PEI, copper(II) can interact both with citric acid and polymer and a ternary compound, where there is one citrate ion per two copper ions, formed at pH > 5.5 [10]. This is highly doubtful for Gd(III), possessing extremely low affinity to amine ligands. Thus, it can be anticipated that the cationic form of PEI could bind a negatively charged gadolinium(III) complex with the maximum possible number of ligand anions. This could provide the best combination of the formation of polymer-ligand hydrogen bonds and polymer-complex electrostatic attraction.

Figure 3 shows the results of NMR relaxation measurements in Gd(III) solutions in aqueous solutions of citric acid and PEI0.

The plots in Figure 3a differ remarkably from the relaxation curves in water for the same concentrations of citric acid. The increase in relaxivity value is 200–300% in a very broad pH range (2–10). A drastic drop of the curve 3 starting from pH 4.25 (Figure 3a) is accompanied by turbidity of the solutions and is presumably caused by the transition of gadolinium compounds into colloidal form. Nevertheless, such solutions also preserve a sufficiently high relaxivity (5–7 mM^−1^s^−1^), which indicates a sufficient accessibility of water molecules for gadolinium ions. Such turbidity of solutions depends on the content of citrate ions and polymer. One example is that the solutions are turbid starting from pH 1.5 at 2 mM PEI, from pH 4.26 at 5 mM, and from pH 8.41 at 10 mM; at 25 mM, the solutions remained transparent in the entire pH range (Figure 3b). It is clear that before pH 6 the minimum PEI content, which provides the formation of a sufficient amount of high-relaxivity gadolinium compounds, corresponds to 5 mM, whereas at 10 mM of PEI, high R_2_ values are limiting.

It is clear from the form of the curves in Figure 3 that there are several types of ternary compounds of gadolinium(III) with citrate ions and PEI. To clarify the metal-to-ligand ratio in the compounds formed in the ternary system, the dependence of relaxivity on the excess content of citrate ions in Gd(III)–PEI solution was analyzed (Figure 4).

With an increase in the excess of the ligand, the plot at pH 4 at the citric acid-to-gadolinium(III) ratio corresponding to 3:1 shows a sharp break and subsequent achievement of the limiting value, the magnitude of which depends on the polymer content R_2_ 46 mM^−1^ s^−1^ (Figure 4a). It is stated from this fact that the compound with the metal-to-ligand ratio of 1:3 is formed quantitatively at pH 4 in the presence of polymer, which is unusual for such acidity of medium. It can be seen that the relaxivity values are not altered at an excess of ligand, which indicates the inability of H_3_L^−^ and H_2_L^2^^−^ anions to exclude the metal complex from the polymer-associated state. At larger pH values (Figure 4b), an increase in the content of ligand is also accompanied by the increase in the relaxivity with the break at the metal-to-ligand ratio close to 1:3 and the relaxivity values become even larger. However, an excess of the ligand larger than three-fold results in a drop of relaxivity, which indicates the decomposition of this compound (Figure 1). It was mentioned above that citrate ions interact with amines with the formation of salt compounds [15,16]. Thus, the effect observed at pH > 5 can be rationalized by a competitive exclusion of the tris-complex of gadolinium preliminarily bound with PEI into water by excess HL^3^^−^ anions, where it is transformed into the [Gd(HL)_2_]^3^^−^ bis-complex, which exists in water under these conditions.

To determine the state of ligand ions in the tris-complex, the curves in Figure 3 recorded for the metal-to-ligand concentration ratio of 1:3 at various amounts of PEI were modeled using the CPESSP program. Because it is difficult to quantify the role of polymer upon interaction of gadolinium(III) with citric acid in aqueous solution of PEI, the apparent equilibrium constants were used upon modeling of the reactions according to Equation (1) (K^app^). Results of mathematical modeling at various PEI contents are given in Table 2.

The curves of relaxation effectiveness vs. pH of the gadolinium(III)–citric acid system (1:3) in the presence of PEI are adequately described by the set of equilibria from Table 1 and Table 2. Thus, there are two types of tris-complexes with the anions of citric acid, which are slightly different by the degree of deprotonation. With an increase in the PEI content, the limiting relaxivity values of the complexes reach a plateau, while the apparent strength of the tris-complex [Gd(H_2_L)_3_]^3^^−^ is practically unchanged. In the second compound, [Gd(H_2_L)_2_HL]^4^^−^, the effect of the polymer content in solution is more pronounced. This fact can be rationalized by the shift of equilibrium to the formation of this ternary compound with an increase in the number of functional groups of polymers that are accessible for attraction.

An increase in the negative charge of the complex with a transition from neutral solutions to weakly basic should enhance the attraction to polymer. However, there are two opposite factors: deprotonation of PEI decreases its ability to attract anions, whereas free citrate ions are transformed into the maximum deprotonated form HL^3^^−^ successfully competing with the [Gd(H_2_L)_2_HL]^4^^−^ complexes, which are similar in charge. This fact can rationalize the decay of relaxivity at pH > 7 (Figure 3b). In this case, as mentioned above, the range of formation of colloidal suspensions is narrowed with an increase in the polymer content and transparent solutions are obtained at the PEI content of 25 mM. There are a range of constant R_2_ values on the corresponding relaxation curve (Figure 3b) at pH 8–9.5. Modeling in the CPESSP program showed that the considered change of the form of relaxation curve at pH 5–10 is described by a mathematical model involving the reactions from Table 1 and Table 2, as well as an additional equilibrium (3):Gd^3+^ + 2H_4_L ⇆ Gd(HL)_2_^3^^−^ + 6H^+^(3)
for which the apparent formation constant is lgK^app^ −12.43 ± 0.08, whereas relaxivity is 20 ± 0.2 mM^−1^s^−1^. At first sight, this complex possesses the same composition as the bis-ligand complex in aqueous medium (No. 8 in Table 1). However, the range of its formation is significantly shifted to the right along pH scale, and relaxivity is larger by 65%. In this bis-complex, Gd(III) possesses less than two sites for the coordination of water molecules or other ligands. Thus, it can be suggested that a strong ternary complex is formed at a large excess of PEI with a decrease in the degree of protonation of polymer in weakly basic solutions, the composition of which can be regarded as Gd(HL)_2_(N)_x_^3^^−^, where (N)_x_ indicates x amino groups of PEIS, which chelate gadolinium. A large excess of polymer (25 mM) is necessary for the formation of such a complex, whereas colloidal precipitates are formed at 10 mM PEI in solution. The suggestion of the formation of the complex with coordinated PEI certainly requires further verification.

### 2.3. Effect of Molecular Mass and Nature of Cationic Polymer on the Complexation of Gadolinium(III) Ions with Citric Acid

The results described above were obtained using PEI with the molecular mass of 60 kDa (PEI0). It was mentioned that the formation of the complexes possessing high relaxivity value is interesting as models of effective contrast agents for MRI. It is also known that toxicity of PEI is inversely proportional to its molecular mass, which in some cases restricts medical and biological potential of the high-molecular-weight specimens. For this reason, the formation of gadolinium citrates was investigated in PEI solutions with various molecular weight values, 25 kDa (PEI1), 1.3 kDa (PEI2), and 0.8 kDa (PEI3); the results of experiments are given in Figure 5.

Colloidal precipitates, which were observed in the case of PEI0, nearly disappeared in the solutions of the Gd(III)–H_4_L–(PEI1, PEI2, or PEI3) systems with the polymer content of 5 mM. In addition, the same high relaxivity values as in PEI0 solutions are achieved only in the case of PEI1 in acidic medium (Figure 5a). Even though relaxivity is higher in acidic solutions with additives of low-molecular PEI2 and PEI3 than those in water, it is lower than 23 mM^−1^s^−1^. With an increase in pH to 8–10, relaxivity in PEI2 and PEI3 solutions amounts to the values for aqueous solutions, indicating the break of bonds of gadolinium citrates with the polymer. The obtained experimental data were processed using the model including the equilibria from Table 1 and Table 2. The calculated values of apparent equilibrium constants of formation of tris-complexes and their relaxivity values in 10 mM solutions of PEI possessing various molecular weights are given in Table 3.

As follows from these data, relaxivity values of the solutions of complexes in the case of PEI3 and PEI2 nearly coincide and are nearly twice as low as the values of R_2_ in PEI1 and PEI0 solutions. At the same time, in spite of the difference in molecular masses of PEI specimens (800–25,000 Da), the K^app^ values nearly coincide for both tris-complexes. Thus, the molecular mass of PEI marginally affects the binding strength of anionic complexes of gadolinium with ammonium groups of the polymer, while the rotation of tris-complexes strongly slows down in the case of high-molecular-weight PEI specimens.

The presence of steps on the plots at pH 8–10 in the solution with 10 mM PEI (Figure 5b) can be caused by the formation of ternary complexes of Gd(HL)_2_(N)_x_^3^^−^ type, which were discussed above for PEI0 (Section 2.2).

Another important problem, which should be addressed, is the role of the nature of functional groups of polymers, which participate in binding with anionic complexes of metal. In this work, this aspect was verified using PDDC, which represents a cationic polyelectrolyte containing quaternized nitrogen atoms. These groups not only preserve positive charge regardless of pH of medium, but also do not form hydrogen bonds.

Figure 6a shows relaxation curves recorded for the solutions of gadolinium(III) citrates with PDDC additive (and PEI0 for comparison). It also shows analogous curves for salt solutions of PDDC and PEI and gadolinium–citrate binary system in water and salt solution. Figure 6b illustrates the determination of the citrate-to-gadolinium ratio in polymer-bound gadolinium citrates at pH 4–8.

Magnetic relaxation behavior of gadolinium citrates in PDDC and PEI0 solutions has some similarities and differences. Firstly, there is an increase in relaxivity at pH 3–7 up to the values intrinsic for low-molecular PEI3 and PEI2 specimens (Figure 5a), even though PDDC possess molecular mass of 100–200 kDa. Secondly, saturation curves at pH 4–6 show the break at the gadolinium-to-citrate ratio of 1:3 indicating the formation of associates of the corresponding composition with polymer. At the same time, the difference, though significant, involves the character of the effect of anions on the stability of these ternary associates. In contrast to PEI, an excess of citric acid up to 10-fold in PDDC solution does not result in the decay of relaxivity on the plots of concentration dependence of relaxivity (plots in Figure 4b and Figure 6b). This fact indicates the absence of competition between tris-complexes and free citrate ions in PDDC solutions. Quaternary nitrogen atoms of the polymer, which cannot form hydrogen bonds with carboxylate groups of citrate ion, represent a possible reason for the absence of bond of citrate ions with head groups of PDDC. However, comparison of the effect of additives of a physiological amount of NaCl ensues in a paradoxical result: the curve for the salt solution of PDDC nearly coincides with the curve of gadolinium citrate in salt solution (Figure 6a). This fact indicates a total destruction of PDDC-bound gadolinium citrates in the presence of 150 mM salt and its transition to water, where they exist as bis-complexes. At the same time, addition of 150 mM of the salt in PEI0 solution only slightly decreases the relaxivity value, indicating the retention of the main amount of gadolinium citrates in the polymer-bound state. Moreover, in salt solution, there is a clearer shoulder band at 20 mM^−1^s^−1^ in the relaxation curve, which reflects the formation of the ternary bis-citrate gadolinium complex discussed above, with amino groups of polymers (Gd(HL)_2_(N)x^3^^−^). The relaxation curve for solutions of Gd:Cit:PEI0:NaCl at pH 2–7 (Figure 7a) was modeled using the described mathematical model involving the reactions from Table 1 and Table 2. Apparent equilibrium constants of formation (lg K^app^) and relaxivity values (R_2_, M^−1^s^−1^) of polymer-bound tris-citrates of gadolinium [Gd(H_2_L)_3_]^3^^−^ and [Gd(H_2_L)_2_HL]^4^^−^ in salt solution of PEI0 corresponded to −7.34 ± 0.10 (43.0 ± 0.8) and −11.38 ± 0.13 (48.1 ± 0.5). Thus, in spite of the fact that tris-citrates remain bound in the polymer globules and, consequently, preserve high relaxivity, their apparent stability decreases compared with that in aqueous solution of PEI, which reflects the competing effect of chloride ions of salt.

To describe the relaxation curve in the Gd:Cit:PDDC system at pH 2–7, the mathematical model including the reactions from Table 1 and Table 2 were used. Apparent equilibrium constants of formation (lg K^app^) and relaxivity (R_2_, M^−1^s^−1^) of polymer-bound tris-citrates of gadolinium [Gd(H_2_L)_3_]^3^^−^ and [Gd(H_2_L)_2_HL]^4^^−^ corresponded to −7.4 ± 0.7 (17.3 ± 0.5) and −11.75 ± 0.24 (20.0 ± 0.4). A large measurement error of the constant of the first compound is rationalized by both its low content under chosen concentration conditions and low difference of relaxivity values of gadolinium aqua-ion and monocitrates [GdH_2_L]^+^, [GdHL]^0^, and [Gd(H_2_L)_3_]^3^^−^(PDDC). The formation of tris-citrate of the second type is also complicated as compared with PEI of any molecular weight. All these results can be interpreted as evidence of the more important role that hydrogen bond formation plays between the groups of ligand and polyelectrolyte, whereas electrostatic attraction is the less important factor upon formation of polymer-bound metal complexes.

### 2.4. Morphology of Gadolinium(III) Complexes with Citric Acid in Polyethyleneimine Solution

It is known that polyethyleneimine macromolecules form the core of micellar aggregate, whereas metal compounds added to the polymer matrix can shape its nanostructure [57,58,59]. In this case, morphology of these particles is influenced by the concentration of metal ions and acidity of medium.

In the previous section, stabilization of the tris-complexes of gadolinium with citric acid by polyethyleneimine was described. Unusually high relaxivity values of these particles (45–50 mM^−1^s^−1^ vs. 4–12 mM^−1^s^−1^ for bis-citrates in water) can only be rationalized by the fact that the mentioned tris-citrates are bound to polymer and appear as large particles, which results in their retarded motion and, consequently, an increase in relaxivity. This statement could be clarified by the direct particle size measurements. For this reason, the techniques which investigate morphology of the compositions of complexes with polymers, were chosen.

Size distribution of polymer-stabilized gadolinium(III) complexes with citric acid depending on the acidity of medium was investigated through DLS and TEM techniques (Figure 7, Table 4). Analysis of TEM images showed that nanostructures of irregular shape with a size of less than 200 nm and their aggregates are formed at pH 4 in the gadolinium(III)–citric acid (Cit)–PEI0 system (Figure 7b), whereas particles of more regular (spherical) shape are formed at pH 6 (Figure 7c,d). The mentioned difference in the shape of nanoobjects is confirmed by the DLS data, since the system is monodisperse: at pH 4 (PdI 0.13) the particle diameter is 70 nm, and at pH 6 (PdI 0.08) the particle diameter corresponds to 100 nm (Figure 7A).

Analysis of the electrophoretic light scattering data (Table 4) of PEI0 solutions, their binary mixtures with Gd(III) or citric acid, and gadolinium(III)–Cit–PEI0 ternary system at two pH values led to next conclusions. The presence of Gd^3+^ ions in a polymer solution insignificantly changes the ZP values, which indicates the absence of interaction of ions with polyelectrolyte in these solutions (confirmed by the relaxation data of solutions, Figure 1).

In a PEI0–citric acid binary system, a drastic increase in the aggregate size and a drastic drop of ZP occurs only at pH 6 (Table 4). In this case, a major fraction of citric acid exists in solution in the form of triple-charged HL^3^^−^ anions, which can crosslink different polymer chains due to the simultaneous effect of several electrostatic and hydrogen bonds. As a result, microparticles are formed, aggregates of which are characterized by low ZP value, and they can be seen by the naked eye. Thus, citrate ions can compete with other anionic particles for binding with polyethyleneimine at pH 6. In a Gd(III)–Cit–PEI0 ternary system, the data of NMR relaxation show the formation of polymer-bound high-relaxivity tris-citrates [Gd(H_2_L)_2_HL]^4^^−^ at this pH, which results in the formation of nearly spherical nanoparticles with the diameter of ca. 100 nm (PDI 0.08). In this case, an excess of citric acid leads to the decrease in relaxivity (Figure 4b), which was rationalized above by their competitive binding with polyelectrolyte.

A notable consequence of the interaction of PEI with gadolinium tris-citrates is an increase in the ZP of associated particles in the ternary system compared with the polymer itself (Table 4). At first glance, since the tris complexes and PEI have opposite charges, their association should have led to a decrease in the ZP, as is the case for the citrate-PEI system at pH 6. Elsewhere was reported [60,61,62] that the degree of PEI protonation at pH 4–6 ranges from 0.3 to 0.89. Thus, it can be assumed that the association with metal complexes (not disturbed by a large excess of chloride ions, Figure 6a) stimulates additional protonation of PEI in the pH range 3–6, which provides both the strength of ternary associates and leads to an increase in the ZP of such nanoparticles at the same pH values (Table 4). Previously the authors of this current study observed a similar effect in the interaction of protonated aminomethylated calixresorcinols with manganese(II)-hydroxyethanediphosphonic acid anionic complexes [31]. Thus, nanosized particles of the Gd (III)–Cit–PEI0 ternary system, which provide an unusually high relaxing ability in solution, are formed due to the crosslinking of polymer chains by tris-citrate gadolinium complexes. In context of the considered possibility of development of a new generation of CAs for MRI, an approach to the preparation of high-relaxivity compositions can be suggested, based on the formation of metal–ligand–polymer ternary associates. The main problem here is to search for the ligands, which combine high binding strength with metal and polymer, and choose components of the composition possessing low toxicity.

## 3. Conclusions

This study has shown that the addition of polyethyleneimine (PEI) to aqueous solutions containing gadolinium(III) and citric acid (H_4_L) resulted in remarkable changes in the measured rates of proton NMR relaxation. In particular, at the concentration ratio of Gd(III):H_4_L = 1:3, there could be 200–300% relaxivity growth in such solutions at pH 3–7 (up to R_2_ 50 mM^−1^s^−1^). Using the CPESSP program, experimental dependence of relaxivity values on the composition of solution and acidity in water and polymer solutions has been described. Two tris-complexes, [Gd(H_2_L)_3_]^3^^−^ and [Gd(H_2_L)_2_(HL)]^4^^−^, have been additionally included into the mathematical model developed according to the results of pH-metric titration and NMR relaxation of gadolinium citrates in water for the solutions with PEI additive. The latter provide a very significant increase in relaxivity as a result of binding with polyelectrolyte. Dependence of the observed increase in relaxivity on the molecular mass of PEI has been analyzed; high relaxivity was achieved only using PEI with the molecular mass of 60 and 25 kDa, while the increase in relaxivity was less than 50% using polymers with the MW of 1.3 and 0.8 kDa. There was a lower effect when using polyelectrolyte with quaternary nitrogen atoms (PDDC), at which the maximum increase in relaxivity was less than 30%. Through transmission electron microscopy (TEM) and dynamic light scattering techniques (DLS), it was determined that particles with the size of 50–100 nm and narrow molecular-mass distribution (PDI 0.08) were formed in solution containing the associate of PEI with tris-ligand complex [Gd(H_2_L)_2_(HL)]^4^^−^. These results are valuable for the optimization of gadolinium ion binding conditions in PEI solutions in the presence of citrate ions during polymer-enhanced ultrafiltration. The established principles of the change of relaxivity in polymer-bound gadolinium citrate solutions are important for the development of the prototypes of high-relaxivity contrast agents for magnetic resonance tomography.

## 4. Materials and Methods

### 4.1. Materials

Gadolinium(III) nitrate hexahydrate, Gd(NO_3_)_3_∙6H_2_O (pure for analysis, Khimreaktiv, Russia), citric acid (ACS reagent, ≥99.5%, Sigma-Aldrich), branched polyethyleneimine, PEI as 50% aqueous solution, (PEIX, where X = 0, 1, 2, 3 for Mw 60 kDa, 25 kDa, 1.3 kDa, and 0.8 kDa, respectively, Fluka), poly(diallyldimethylammonium chloride), PDDC as 35% aqueous solution (Sigma-Aldrich), sodium chloride (chemically pure, Khimreaktiv, Russia), sodium hydroxide, nitric and hydrochloride acids (pure for analysis, Khimreaktiv, Russia) were used. Ultra-purified water (18.2 MΩcm resistivity at 25 °C) was produced from Direct-Q 5 UV equipment (Millipore S.A.S. 67, 120 Molsheim, France). Experiments and measurements were conducted at 298 K. The temperature was maintained using a Haake DC10 (Thermo Electron) cryo thermostat. Polymer concentrations are expressed in their “monomeric” units relative to their respective molecular weights.

A “Starter 3100” (Ohaus) pH-meter was used for measuring acidity values of solutions, and was calibrated using standard buffers (pH 4.01, 7.00 and 9.00); pH metric titration was performed using an A-600 autotitrator (Kyoto) with carbonate free alkali.

### 4.2. NMR Relaxation

Spin-spin (transverse) T_2_ relaxation times were measured using pulsed NMR relaxometer Minispec MQ20 (Bruker) with an operational frequency of 19.65 MHz by applying a standard Carr-Purcell pulse sequence modified by Meiboom and Gill with a measuring accuracy error better than 3%. The experimentally measured transverse relaxation times (T_2_)_obs_, s, were inverted into the relaxation rates (1/T_2_)_obs_, s^−1^. The relaxation rate is the sum of the two main contributions: the relaxation of protons in water (1/T_2_)_d_ (diamagnetic component) and the relaxation of the protons around the paramagnetic ion (1/T_2_)_p_ (paramagnetic component):(4)$${\left(\frac{1}{T_{1,2}}\right)}_{obs}={\left(\frac{1}{T_{1,2}}\right)}_{p}+{\left(\frac{1}{T_{1,2}}\right)}_{d}$$

The paramagnetic component, (1/T_2_)_p_, was calculated according to Equation (4) as the difference between the measured relaxation rate (1/T_2_)_obs_ (measured for Gd-containing solutions) and the diamagnetic component (1/T_2_)_d_ (the same solutions but without gadolinium ions; for diluted aqueous solutions this is equal to 0.4 s^−1^).

Using gadolinium concentration, C (mM), the paramagnetic component, (1/T_2_)_p_, was converted into relaxivity R_2_, mM^−1^s^−1^, according to Equation (5):(5)$$R_{2}={\left(\frac{1}{C_{M}T_{2p}}\right)}_{}$$

Provided that several complexes of a paramagnetic metal exist in a solution, the values of the paramagnetic contribution to relaxation R_2_ are averaged by taking into account the contributions of all *i* complexes present in the solution:(6)$$R_{2}=\overset{N}{\underset{i=1}{\sum }}\alpha_{i}\cdot R_{2i}$$

This additivity was taken into account and is the basis for using a computer simulation of experimental data in determining the composition and stability of existing metal complexes. 

### 4.3. Modeling of Equilibria

The values of the dissociation constants of citric acid and the equilibrium of complexation were obtained using mathematical models of the studied systems, including the equilibrium schemes (with stoichiometric coefficients for the reagents), the values of the equilibrium formation constants of complexes and their relaxivities. In polymeric solutions the same models were applied with the constants and relaxivities as for aqueous solution, and new complexes where added, the composition of which was proved by fitting the experimental relaxation data. For such new complexes the revealed equilibrium constants are apparent, as they do not account directly for participation of the polymer in binding. The optimization of the numerical parameters (constants and relaxivities) was carried out using the CPESSP computer program (the foundations of the program are outlined in [63] with assessment of the reliability by the Fisher criterion.

### 4.4. Size and Morphology

The mean particle size, zeta potential and polydispersity index were determined by dynamic light scattering (DLS), using the Malvern Instrument Zetasizer Nano (Worcestershire, UK). The measured autocorrelation functions were analyzed by Malvern DTS software, applying the second-order cumulant expansion methods. The size (hydrodynamic diameter, nm) was calculated according to the Einstein-Stokes relationship D = kBT/3πηx, in which D is the diffusion coefficient, kB the Boltzmann’s constant, T the absolute temperature, η the viscosity, and x the hydrodynamic diameter of nanoparticles. The diffusion coefficient was determined at least in triplicate for each sample. The average error of measurements was approximately 4%. All samples were diluted with ultra-purified water to suitable concentration and analyzed in triplicate.

Transmission electron microscopy (TEM) images were obtained using a Hitachi HT7700 Exalens microscope, Japan. The images were acquired at an accelerating voltage of 100 keV. Samples were dispersed on 300 mesh 3 mm copper grids (Ted Pella) with continuous carbon formvar support films.

## Figures and Tables

**Figure 1 ijms-22-11590-f001:**
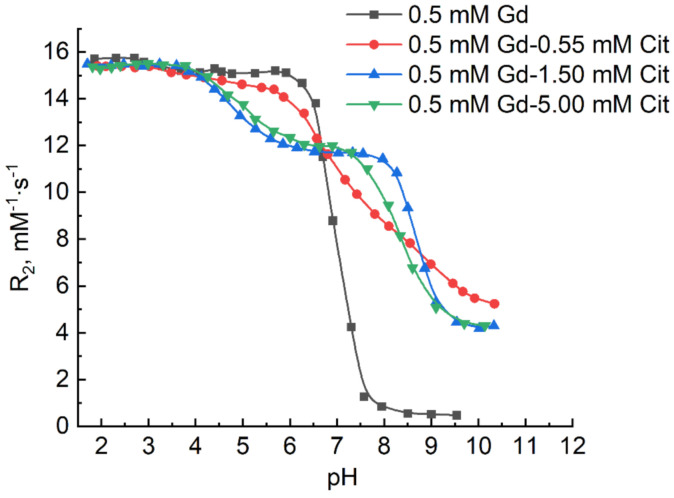
Variation of R_2_ values at different pH values of the solutions of Gd^3+^ ions and its complexes with citric acid in water. C_Gd(III)_ 0.5 mM, C_H4L_ 0–5 mM, C_PEI_ 10 mM.

**Figure 2 ijms-22-11590-f002:**
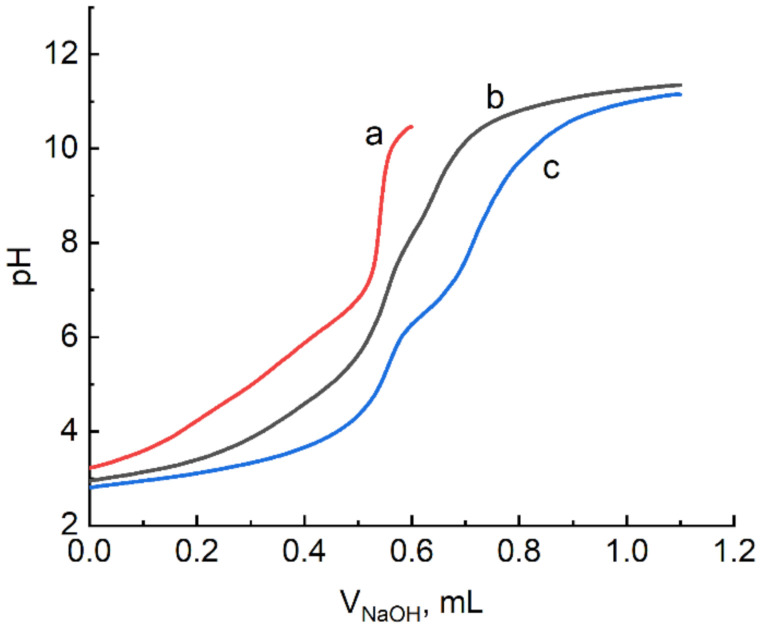
Titration curves of citric acid solution in (**a**) water and (**b**,**c**) in the presence of gadolinium ions. C_H4L_ 1.0 mM, C_NaOH_ 0.312 M, C_Gd(III)_ (**b**) 0.5 mM, (**c**) 1.0 mM.

**Figure 3 ijms-22-11590-f003:**
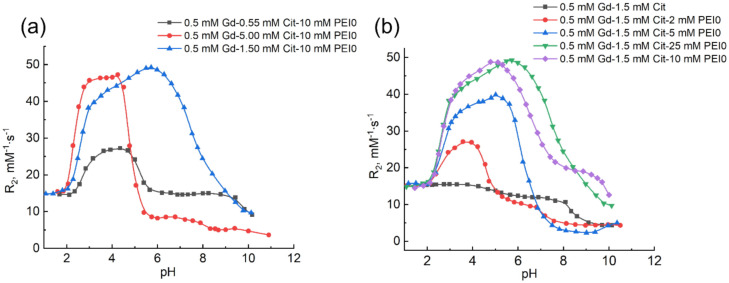
Dependence of the relaxation effectiveness of the solutions containing Gd^3+^ ions, PEI0, and citric acid on the acidity of medium at various contents of (**a**) citric acid and (**b**) PEI. C_Gd(III)_ 0.5 mM, C_PEI_ 0–25 mM, C_H4L_ 0–5 mM.

**Figure 4 ijms-22-11590-f004:**
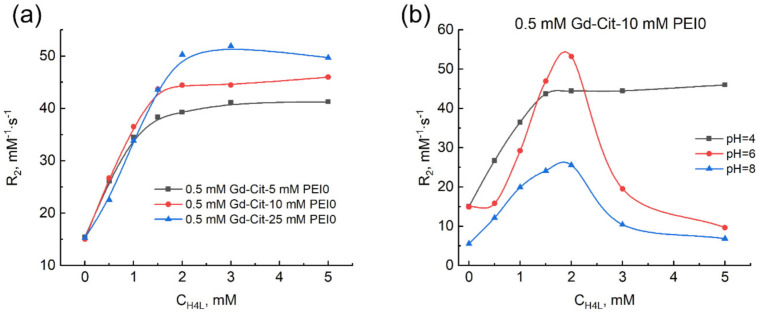
Dependence of the relaxation effectiveness in the gadolinium(III)–citric acid–PEI0 system on the concentration of ligand, C_H4L_. C_Gd(III)_ = 0.5 mM, C_PEI_ = 5.0 mM (**a**), 10 mM (**a**,**b**), and 25 mM (**a**). pH 4 (**a**,**b**), pH 6 (**b**), pH 8 (**b**).

**Figure 5 ijms-22-11590-f005:**
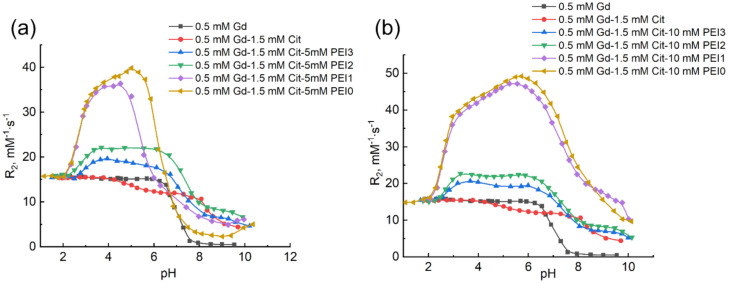
Dependence of relaxation effectiveness of the solutions containing Gd^3+^ ions, citric acid, and polymer on acidity of medium at the constant content of citric acid in the presence of PEI (PEI0, PEI1, PEI2, or PEI3). C_Gd(III)_ 0.5 mM, C_H4L_ 1.5 mM, C_PEI_ 5 mM (**a**) 10 mM (**b**).

**Figure 6 ijms-22-11590-f006:**
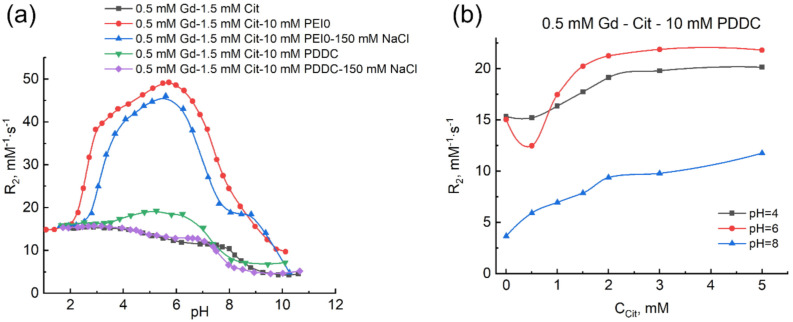
Dependence of relaxation effectiveness of solutions containing Gd^3+^ ions and citric acid in the presence of PEI0 and PDDC on the acidity of medium (**a**) and on the concentration of citric acid in presence of 10 mM PDDC (**b**). C_Gd(III)_ 0.5 mM, C_H4L_ 1.5 mM, C_PEI_ 5 mM (**a**) 10 mM (**b**), C_PDDC_ 10 mM (**b**), C_NaCl_ 150 mM (**a**).

**Figure 7 ijms-22-11590-f007:**
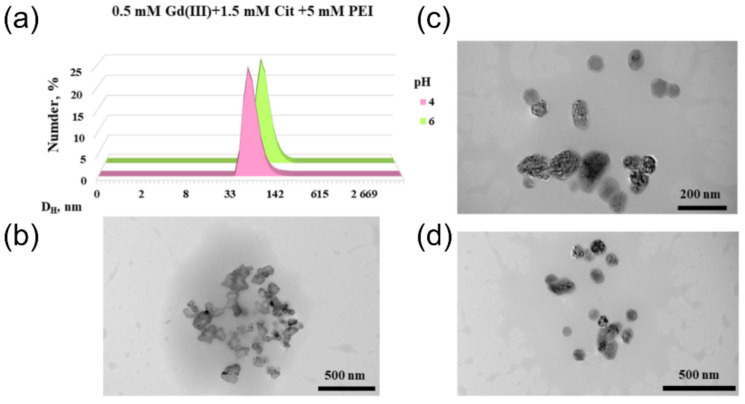
Size distribution characterized by DLS (**a**) and TEM images (**b**–**d**) of Gd(III)–Cit–PEI0 system for pH 4 (**b**), pH 6 (**c**,**d**).

**Table 1 ijms-22-11590-t001:** Dissociation constants of citric acid, equilibrium constants of formation of citrate complexes of Gd(III), and their stability constants and relaxivity values.

No.	Equilibrium	lgK	pK_a_	lgβ	R_2_, mM^−1^s^−1^
1	H_4_L ⇆ H_3_L^−^ + H^+^	−3.09 ± 0.01	3.09		-
2	H_4_L ⇆ H_2_L^2^^−^ + H^+^	−7.78 ± 0.01	4.69		-
3	H_4_L ⇆ HL^3^^−^ + H^+^	−13.95 ± 0.02	6.17		-
4	Gd^3+^ + H_4_L ⇆ GdH_2_L^+^ + 2H^+^	−1.90 ± 0.03		5.88	(15.5)
5	Gd^3+^ + H_4_L ⇆ GdHL^0^ + 3H^+^	−5.51 ± 0.02		8.44	(15)
6	Gd^3+^ + H_4_L ⇆ GdL^−^ + 4H^+^	−12.31 ± 0.04		15.64	7.2 ± 0.2
7	3Gd^3+^ + 3H_4_L ⇆ Gd_3_L_3_H_-2_^5^^−^ + 14H^+^	−50.14 ± 0.16		19.71	3.8 ± 0.4
8	Gd^3+^ + 2H_4_L ⇆ Gd(HL)_2_^3^^−^ + 6H^+^	−15.15 ± 0.08		12.75	12.2 ± 0.2
9	Gd^3+^ + 2H_4_L ⇆ GdL_2_^5^^−^ + 8H^+^	−32.25 ± 0.14		23.65	4 ± 0.1

**Table 2 ijms-22-11590-t002:** Equilibria of the reactions of formation of the complex forms in the gadolinium(III)–citric acid (H_4_L)–PEI0 system, their magnetic-relaxation characteristics, and apparent equilibrium constants (K^app^).

Reaction of Formation	lgK^app^ (R_2_, mM^−1^s^−1^)		
	5 mM PEI	10 mM PEI	25 mM PEI
Gd^3+^ + 3H_4_L ⇆ Gd(H_2_L)_3_^3^^−^ + 6H^+^	−5.71 ± 0.07 (38 ± 0.2)	−5.48 ± 0.15 (44.6 ± 0.7)	−5.68 ± 0.14 (44.8 ± 0.9)
Gd^3+^ + 3H_4_L ⇆ Gd(H_2_L)_2_HL^4^^−^ + 7H^+^	−10.02 ± 0.22 (40.3 ± 0.3)	−10.01 ± 0.12 (48.5 ± 0.6)	−9.47 ± 0.10 (49.6 ± 0.6)

**Table 3 ijms-22-11590-t003:** Equilibria of formation reactions of gadolinium(III) tris-citrates formed in solutions of PEI of different molecular weight, their relaxivity, and the values of the apparent equilibrium constants (K^app^).

Reaction of Formation	lgK^app^ (R_2_, mM^−1^s^−1^)		
	PEI3	PEI2	PEI1
Gd^3+^ + 3H_4_L ⇆ Gd(H_2_L)_3_^3^^−^ + 6H^+^	−5.94 ± 0.21 (22.5 ± 0.2)	−5.97 ± 0.26 (23.0 ± 0.3)	−5.71 ± 0.09 (44.8 ± 0.4)
Gd^3+^ + 3H_4_L ⇆ Gd(H_2_L)_2_HL^4^^−^ + 7H^+^	−10.98 ± 0.15 (22.6 ± 0.3)	−11.17 ± 0.12 (22.5 ± 0.3)	−11.01 ± 0.10 (49.8 ± 0.5)

**Table 4 ijms-22-11590-t004:** Zeta potential (ZP) of investigated Gd(III)–Citrate–PEI0 compositions.

Composition	ZP (mV)	
	pH 4	pH 6
5 mM PEI0	+17 ± 1	+14 ± 1
1.5 mM Gd(III) + 5 mM PEI0	+19 ± 1	+13 ± 1
1.5 mM Cit + 5 mM PEI0	+17 ± 1	+4 ± 0.4
0.5 mM Gd(III) + 1.5 mM Cit + 5 mM PEI0	+29 ± 0.3	+18 ± 1

## Data Availability

The data presented in this study are contained within the article or in Appendix A, or are available on request from the corresponding authors: Evgenia Burilova, Rustem Amirov.

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
