# Peer review of "Design of High-Relaxivity Polyelectrolyte Nanocapsules Based on Citrate Complexes of Gadolinium(III) of Unusual Composition"

_ijms, 2021, doi:10.3390/ijms222111590_

Round 1

Reviewer 1 Report

The manuscript entitled " Design of high-relaxivity polyelectrolyte nanocapsules based on citrate complexes of gadolinium(III) of unusual composition " submitted by Burilova 1 et al. describes the details of complexation of gadolinium(III) ions with citric acid (H4L) in water and aqueous solutions of cationic polyelectrolytes. The manuscript presents an impressive degree of collaboration drawn from experts in synthesis, NMR spectroscopy and calculations. This work would appeal to a wide range of readership. Publication is recommended.

Author Response

Thank you for your review

Reviewer 2 Report

In the present experimental study supervised by E. Burilova and R. Amirov, the authors investigate the large increase of transverse relaxivity  in water of dissolved citrate complexes of the paramagnetic Gd(III) cation due to the binding interactions of some of these complexes with branched cationic polymers polyaziridine or poly(ethylene imine) (PEI) of rather large molecular weights  25 kDa. The relaxivity increase, of up to a factor of three with respect to the aqua ion , markedly depends on the pH of the solution, on the relative concentrations of Gd(III), citric acid, and PEI, and on the PEI molecular weight. The wealth of the relaxivity data, the large size of the observed effects, and the importance of the molecular understanding of the interactions of metal complexes with polymers, both for fundamental knowledge and for applications in ecology with polymer-enhanced ultrafiltration (PEUF) and in medicine with the design of more efficient MRI contrast agents, clearly pleads for publication in Int. J. Mol. Sci.. However, prior to publication, various theoretical and pedagogical issues should be solved and wording points need to be clarified.

Theoretical issues:

1) The transverse relaxivity of a paramagnetic complex of concentration  [M], usually denoted as  [s-1·mM-1], is the slope of the transverse relaxation rate  at small concentration of the complex. When the complexes interact with each other or with other entities, the dependence of  vs  can significantly deviate from a linear law, even at small values of  [1]. Then,  is only an effective relaxivity which can depend on  and on the composition of the solution. This issue is particularly relevant in the present context and should be pointed out. Besides, the usual notation  should be employed rather than .

2) The likely general theoretical expression , the relaxivity parameters of which are fitted to reproduce the experimental values , should be given in the main text. The nature of the continuous curves displayed in the figures should be defined. Do these curves represent the fitted  values or are they just used to link the measured values? The fitted  curves should be displayed unless they nearly pass through all measured values to within the experimental errors.

3) The molecular rationale of the interpretation of the observed transverse relaxation rate values  is based on the number  of water molecules coordinated to the Gd(III) ion in each citrate complex  and on the rotational correlation time  of this complex. This rotational correlation time is roughly proportional to the volume  of the complex and given by , where  is the viscosity of the solution. Though the complicated effects of the electronic spin relaxation of Gd(III) on  can still be significant at the Minispec field of about 20 MHz, the authors can reasonably neglect them in their experimental approach and use the high-field approximation given by Eqs. (1), (2), (18) of Ref. [1]. Without going into theoretical subtleties, the authors are invited to check that the fitted relaxivities  are very roughly compatible with the expected numbers  and rotational correlation times  of their various citrate complexes, in particular for the tris-citrate associates that bind to PEI particles of large sizes and long correlation times.

4) The cationic nature of PEI is invoked, but the pKas of its N-H groups are not used. The authors could quote the paper by Curtis et al. [2] and perhaps other complementary studies and consider how this information can help them interpret their data.

5) lines 397-399: could you propose likely molecular models of H-bond formation?

6) Table 4: The three definitions of hydrodynamic diameters and their very different estimates should be discussed since they seem to lead to opposite conclusions. Can the information conveyed by the Zeta potential be better exploited?

Pedagogical issues:

1) The information provided by the figures is often difficult to grasp because the small symbols and colors are not different enough and do not seem to obey any logical requirement. For instance, the experimental values  and their theoretical counterparts for the aqua ion could be represented by circles and dotted curves, the analogous properties for the Gd(III) solutions with citric acid could be displayed by triangles and dashed curves, while  the analogous properties for the Gd(III) solutions with citric acid and PEI could be differentiated by squares and continuous curves. Of course, very different colors should be employed to distinguish between concentrations. Parts A and B of the figures should be indicated. When necessary, additional parts could be added in order to limit the overflow of information per figure.

Wording points:

line 64: an informative

line 80: 4,5-dihydroxybenzene

line 85: the formation of which

line 98: GdDOTA is very safe, GdEDTA will probably never be used in medicine

line 108: transverse NMR relaxation at 19.65 MHz

line 124: indicate the fact that the Gd hydroxo complexes are insoluble

line 176: EDTA and DTPA do not saturate the coordination environment of Gd

line 201: proton abstraction? not English

line 236: 5 mM of what? 10 mM of what? Please, explain.

line 277: alters? Please, explain.

lines 356-357: the notation CA for concentration of A is unusual

line 372: paradoxical

lines 378-379: reflects, not responsible for

[1] Bonnet et al., A Rigorous Framework To Interpret Water Relaxivity. The Case Study of a Gd(III) Complex with an r-Cyclodextrin Derivative. JACS 2008, 130, 10401-10413.

[2] Curtis et al., Unusual Salt and pH Induced Changes in Polyethylenimine Solutions. PlOS ONE 2016, 1-20.

Author Response

 Theoretical issues:

Q1

Reply:

Indeed, provided that several complexes of a paramagnetic metal exist in a solution, the values of the paramagnetic contribution to relaxation R2 = (R2obs - R20)/cpara are averaged due to the additive addition of contributions from all Ai complexes present in the solution:

This additivity was taken into account by us and is the basis for using computer simulation of experimental data in determining the composition and stability of existing metal complexes.

As for the change in the slope of R2 from linearity with an increase in the content of metal ions (cpara) while maintaining a sufficient and constant excess of the ligand and at a constant acidity of the medium, it usually takes place when the formation of polynuclear complexes is possible (the simplest case is dimerization of the type 2ML = M2L2). According to our experience with citrate complexes, such processes are indeed possible for gadolinium, but at a higher content of its ions than was used in this work (0.5 mM) and usually after pH 7.

Q2

Reply:

The specified equation is inserted into the main text.

The lines on the graphs are not the result of simulations, but serve only to illustrate the trend in the experimental points.

The fitted R2 curves together with the complex distribution diagrams in all simulated systems are presented in the Supplementary.

Q3.

Reply:

Unfortunately, relaxivity sometimes turns out to be weakly sensitive to the substitution of water molecules in the first sphere of the paramagnetic cation, while the size of the complex does not increase so much as to affect the rate of its rotation. Even in this work, we are faced with such an effect for the first two monoligand complexes GdH2L and GdHL, the relaxivity of which differs little from that of gadolinium aqua ions.

The relationship between relaxivity and solution viscosity for ions in which τr controls relaxation is not obvious. Back in the 1960s, attempts to influence the rotation of ions through the total viscosity of solutions, regulated by the addition of glycerol or salts, did not materialize. Also, however, there is no direct relationship between relaxation and the molecular weight of the complex.

For example, the authors of the work [Montembault V., Soutif J.-C., Brosse J.-C. Synthesis of chelating molecules as agents for magnetic resonance imaging. Polycondensation of diethylenetriaminepentaacetic acid bisanhydride with diols and diamines// React. Polym. 1996. V. 29, N 1. P. 29-39.] did not find any changes in the relaxivity of the gadolinium complex with a polymer ligand containing DTPA fragments in comparison with the gadolinium-DTPA complex. In other cases, the relaxivity reaches the limit and practically does not change with an increase in the molecular weight.

In this regard, we did not try to find a quantitative relationship between the observed relaxation of gadolinium tris-citrate solutions associated with PEI and the sizes of these associates.

Q4

Reply:

The article by Curtis et al. referred to is about linear PEI, which contains the same type of secondary nitrogen atoms. The PEI used in our work is branched and contains primary, secondary and tertiary nitrogen atoms, and in different proportions. For branched PEI, attempts have been made to quantitatively describe its acid-base properties; for example, in the works (the last three are included to the reference list of the revised manuscript):

  1. Burgess, R. R. Use of polyethyleneimine in purification of DNA-binding proteins / In: Methods in Enzymology. Robert T. Sauer (Ed.). - New York: Academic Press, 1991. - P. 3-10.
  2. Horn, D. Polyethyleneimine - physicochemical properties and applications / In: Polymeric Amines and Ammonium Salts. Goethals, E. J. (Ed.) - Oxford: Pergamon Press, 1979. - P. 333-355.
  3. Suh, H.J. Paik, B.K. Hwang. Ionization of poly(ethyleneimine) and poly(allylamine) at various pH’s / Bioorg. Chem. - 1994. - V. 22. - P. 318-327.
  4. Borkovec, G.J.M. Koper. Proton binding characteristics of branched polyelectrolytes // Macromolecules. - 1997. - V. 30, N 7. - P. 2151-2158.
  5. von Harpe, H. Petersen, Y. Li, T. Kissel. Characterization of commercially available and synthesized polyethylenimines for gene delivery. Journal of Controlled Release 69 (2000) 309–322. DOI: 10.1016/s0168-3659(00)00317-5.
  6. Li, S. M. Ghoreishi, J. Warr, D. M. Bloor, J. F. Holzwarth, E. Wyn-Jones. Binding of Sodium Dodecyl Sulfate to Some Polyethyleneimines and Their Ethoxylated Derivatives at Different pH Values. Electromotive Force and Microcalorimetry Studies. Langmuir 2000, 16, 3093-3100. DOI:10.1021/la9910172
  7. Ya. Zakharova, F.G. Valeeva, D.B. Kudryavtsev, A.V. Bilalov, A.Ya. Tret´yakova, L.A. Kudryavtseva, A. I. Konovalov, V. P. Barabanov. Sodium dodecyl sulfate-polyethyleneimine-water system. Self-organization and catalytic activity. Russ. Chem. Bull. 2005. N3. P. 641-649.

These works give an approximate estimate of 70% of the degree of protonation of PEI at pH 5.5 and 55% at pH 7 or even 0.3-0.89 at pH 4-6. Based on this, we assume that the ability of PEI to associate with gadolinium tris-citrates is achieved in strongly acidic solutions, where the protonation of PEI is maximal, and after pH 7 this ability practically disappears.

The reasons why the ZP of the Gd-Cit-PEI associates increases compared to the ZP of the PEI itself are discussed in the revised manuscript.

Q5

Reply:

Based on the known crystallographic data on the structure of gadolinium bis-citrates, we assume the following structure of tris-citrates:

и              

                   [Gd(H2L)3]3-                                                                  [Gd(H2L)2(HL)]4-

In both complexes, three dissociated carboxy groups are located on the periphery of the complex, and undissociated carboxyl groups are also located there. All of them can easily form hydrogen bonds with the ammonium groups of the polymer, which, due to the branching of its skeleton, can surround such a complex without much distortion. Such schemes are rather speculative, therefore we do not present them in the text of the manuscript.

Q6

Reply:

The hydrodynamic diameters obtained by the DLS method for single (PEI) and double systems (PEI-metal and PEI-citric acid) have a high polydispersity index. In this regard, in the new version of the article we do not present them in Table 4. The discussion of the results of the DLS method is carried out, giving the hydrodynamic diameter for the ternary system (PEI-metal-citric acid) in Fig. 7a, and the zeta potential of all systems in Table 4.

Pedagogical issues:

Reply:

We have tried our best to correct the graphics on the figures according to the Referee's comments.

Wording points:

Reply:

line 64: an informative

corrected

line 80: 4,5-dihydroxybenzene

corrected

line 85: the formation of which

corrected

line 98: GdDOTA is very safe, GdEDTA will probably never be used in medicine

corrected

line 108: transverse NMR relaxation at 19.65 MHz

corrected

line 124: indicate the fact that the Gd hydroxo complexes are insoluble

Indicated

line 176: EDTA and DTPA do not saturate the coordination environment of Gd

corrected

line 201: proton abstraction? not English

corrected

line 236: 5 mM of what? 10 mM of what? Please, explain.

corrected

line 277: alters? Please, explain.

corrected

lines 356-357: the notation CA for concentration of A is unusual

corrected

line 372: paradoxical

corrected

lines 378-379: reflects, not responsible for

corrected

Reviewer 3 Report

The work of Burilova et al. is devoted to study of complexation of gadolinium(III) ions with citric acid in aqueous solutions of cationic polyelectrolytes, such as poly(ethylene imine) PEI and polydiallyldimethylammonium chloride PDDC, by nuclear magnetic relaxation and pH-metry. Authors demonstrate, that addition of PEI to Gd(III)/citric acid solution results in changes in rates of proton NMR relaxation, which is rationalized as a result of binding of Gd(III) tris-complexes with PEI. Dependence of increase of relaxivity on molecular mass of PEI and substitution of PEI on PDDC was also studied. The particles, which are formed in a solution of PEI and Gd(III) complexes, were studied by TEM and DLS techniques. The work describes the interaction in the system Gd(III)/(citric acid)/PEI quite fully and variably and can be published in Int. J. Mol. Sci.

Minor questions:

The introduction looks too long, it is recommended to shorten its length by transferring some of the material to the discussion section.

The fig.2-6 do not include the letter designations (a), b) etc) given in the figure captions, they should be added.

The caption to the figure 6 is unclear, it should be worded differently.

Table 4. Should it be 5 mM PEI0 instead of 5 mM PEI?

Reference [60] should be given in full, with the title of the work.

Author Response

Minor questions

Reply

The introduction looks too long, it is recommended to shorten its length by transferring some of the material to the discussion section.

in our opinion, all aspects of the work are briefly reflected in the introduction, and we tried to discuss the known approaches both from the point of view of PEUF and the possibilities of contrasting MRI images.

The fig.2-6 do not include the letter designations (a), b) etc) given in the figure captions, they should be added

corrected

The caption to the figure 6 is unclear, it should be worded differently

corrected

Table 4. Should it be 5 mM PEI0 instead of 5 mM PEI?

corrected

Reference [60] should be given in full, with the title of the work

corrected
